



# GlobSim (v1.0): Deriving meteorological time series for point locations from multiple global reanalyses

Bin Cao, Xiaojing Quan, Nicholas Brown, Emilie Stewart-Jones, and Stephan Gruber

Department of Geography and Environmental Studies, Carleton University, Ottawa, Canada

**Correspondence:** Stephan Gruber (stephan.gruber@carleton.ca)

**Abstract.** Simulations of land-surface processes and phenomena often require driving time series of meteorological variables. Corresponding observations, however, are unavailable in most locations, even more so, when considering the duration, continuity and data quality required. Atmospheric reanalyses provide global coverage of relevant meteorological variables, but their use is largely restricted to grid-based studies. This is because technical challenges limit the ease with which reanalysis data can

be applied to models at the site scale. We present the software toolkit GlobSim, which automates the downloading, interpolation and scaling of different reanalyses—currently ERA-5, ERA-Interim, JRA-55 and MERRA-2—to produce meteorological time series for user-defined point locations. The resulting data have consistent structure and units to efficiently support ensemble simulation. The utility of GlobSim is demonstrated using an application in permafrost research. We perform ensemble simulations of ground-surface temperature for ten terrain types in a remote tundra area in northern Canada and compare the

results with observations. Simulation results reproduced seasonal cycles and variation between terrain types well, demonstrating that GlobSim can support efficient land-surface simulations. Ensemble means often yielded better accuracy than individual simulation and ensemble ranges additionally provide indications of uncertainty arising from uncertain input. By improving the usability of reanalyses for research requiring time-series of climate variables for point locations, GlobSim can enable a wide range of simulation studies and model evaluations that previously were impeded by technical hurdles in obtaining suitable

data.

## 1 Introduction

Models that represent the interactions between the land surface and the atmosphere are often used to investigate biogeochemical, cryospheric and hydrologic phenomena. Because they require meteorological forcing—with daily or finer temporal resolution, for extended periods and without gaps—site-specific applications such as process studies or model testing are lim-

ited to few locations where high quality ground observations are available. In this context, global atmospheric reanalyses can substitute for lacking observations or supplement incomplete records. They assimilate a broad range of observations into numerical weather-prediction models, usually have coarse grid spacing (10–100 km) and are often used for large-area studies in atmospheric and hydrological modelling (e.g. Žagar et al., 2018; Albergel et al., 2018). Their application to point locations (e.g., Ekici et al., 2015; Westermann et al., 2016), however, is currently limited, likely because accessing data is technically

involved. The software GlobSim, which is presented here, aims to contribute to overcoming this obstacle.



The suitability of reanalysis data for individual projects depends on the environment studied, the skill of the reanalysis in representing it and characteristics of the intended application. Several global reanalysis products are available to drive simulations or ensembles of simulations. Their relative suitability for specific simulation studies and locations is likely to vary because they rely on different assumptions, parameterizations or assimilated input data, and these differences result in biases

that are spatially heterogeneous and specific to one or several meteorological variables (Decker et al., 2012; Zhang et al., 2016). The use of model ensembles is established to evaluate uncertainty or improve predictive accuracy in simulation studies (e.g., Tebaldi and Knutti, 2007). Ensembles consist of multiple simulations for a given location and time that that differ in one or more ways. For instance, they can be generated by varying the forcing data (e.g. Guo et al., 2018), the models themselves (e.g. Westermann et al., 2015b) or the structure and parameters or a model (e.g. Gubler, 2013). Because meteorological time series

that drive models are a major source of uncertainty, having multiple reanalysis products available for ensemble simulations is an important step in estimating and reducing overall simulation uncertainty.

Four main challenges impede the use of reanalysis data for simulating point locations. (i) Technical delivery: Available data is well documented but its structure largely reflects the needs and conventions common in atmospheric simulation and, as a consequence, individuals used to working with data from meteorological stations may find their handling difficult. For

example, ERA-Interim (ERAI) provides certain variables as accumulations over time intervals and these must be disaggregated if instantaneous values are desired. (ii) Differences between reanalyses: Reanalyses differ in their spatial and temporal grids as well as the conventions and units used for their variables and files (cf. Arsenault et al., 2018). (iii) Spatial scale: The coarse-grid reanalysis data requires spatial interpolation to the locations of interest and heterogeneous environments such as mountains or coasts may require additional, subsequent scaling procedures (e.g., Fiddes and Gruber, 2014; Sen Gupta and Tarboton, 2016;

Cao et al., 2017). (iv) Differences between reanalysis and model: Reanalysis data usually require unit conversion, computation of derived variables (e.g., wind direction from northward and eastward wind speeds) and temporal interpolation in order to be suitable for driving specific models. Although addressing these four main challenges is not conceptually difficult, it does represent a technical hurdle that must be overcome in order to more fully materialize the benefits of reanalysis data for driving models at point locations.

This contribution describes the software GlobSim, named as a portmanteau of "Global Simulator", which produces time series from multiple reanalyses for driving model simulation at point locations. As a demonstration, we apply GlobSim to ground-surface temperature (GST) simulation in a densely instrumented and well described location in a tundra environment near Lac de Gras, Northwest Territories, Canada. This demonstrator is relevant for investigating permafrost and its changes, one example of a research field where the lack of data to drive simulations is particularly severe. The objectives of this study

are (1) to describe the software GlobSim and test its results for blunders, and (2) to quantify the performance of simulations supported by GlobSim in a demonstrator application. For this, we compare ensemble members and means to statistical summaries of observations for different terrain types. Because reanalyses are imperfect and their performance can vary regionally and temporally (Decker et al., 2012; Fiddes and Gruber, 2014), the demonstrator can inspire, but not quantitatively underpin, applications in other areas, at other times and using other models. The study area has been chosen to test whether the

combination of coarse-scale reanalysis data with fine-scale information on surface and subsurface characteristics (the terrain





types) can reproduce the seasonal temperature cycles observed and the resulting fine-scale differentiation of ground temperature. This is relevant to potential users of atmospheric simulation data and also, for the development of atmospheric models as it can inform decisions related to the trade-off between increased resolution in the atmosphere or increased tiling (sub-grid) resolution of their land-surface components. Finally, improved simulation of land-surface processes and phenomena will be-

come increasingly important for supporting decision making under climate change because it can help to estimate likely future environmental conditions. In this context, flexible model evaluation and application globally is an important first step.

## 2   Background

### 2.1   Downscaling of reanalyses

Most global atmospheric models produce output at coarse spatial scales (10–200 km) and for many applications, these data

need to be downscaled, a process that has been described as making the link between the state of variables representing a large space and the state of variables representing a much smaller space (Benestad et al., 2008). For this, two main approaches exist: dynamical downscaling (e.g., Bieniek et al., 2016), which relies on nested atmospheric models with increasingly fine resolution and decreasing spatial extent, and empirical-statistical downscaling (e.g., Daly et al., 2008), which employs observations to derive mapping functions for linking coarse and fine scales. As GlobSim is intended to operate in areas without observations,

this section emphasizes empirical-statistical downscaling methods that are tolerant to application far away from the observation with which they were derived.

Several empirical-statistical methods exist to downscale gridded meteorological data or to spatialize observations in hetero-geneous environments. These applications are different but, especially when aiming to perform downscaling in areas without observations, show significant overlap. For example, Hungerford et al. (1989); Liston and Elder (2006) and Thornton et al.

(2012) produced spatial data by interpolating observations based on topo-climatic variables (e.g., elevation, slope angle and slope aspect) and vegetation, and Daly et al. (2008) produced grids of mean monthly precipitation and temperature with a method that is frequently used in application studies (e.g., Jafarov et al., 2012). More recently, lapse rates for adjusting surface air temperature to fine-scale topography were derived from reanalysis pressure-level data (Gruber, 2012; Fiddes and Gruber, 2014). This allows lapse rates to vary temporally and physically consistent with the atmospheric conditions. Cao et al. (2017)

further developed the surface air temperature downscaling by parameterizing fine-scale inversions such as cold air pooling. Downscaling methods for other variables such as shortwave and longwave radiation, precipitation and wind speed suitable for mountains exist (Fiddes and Gruber, 2014; Sen Gupta and Tarboton, 2016) and can inform application also in gently slop-ing terrain and the potential of these scaling methods has been demonstrated in simulation studies (e.g., Fiddes et al., 2015; Westermann et al., 2015a).





## 2.2 Ensemble simulation

Model uncertainty arises from input data and the models themselves (Gupta et al., 2005; Gubler et al., 2013) but the quantitative evaluation of this uncertainty is difficult when models are complex (Murphy et al., 2004). This is also true when simulations of land-surface processes or phenomena are forced by reanalyses because their uncertainty—arising from e.g., the observational

data, assumptions, model structure, initialization and parameters used—propagate into the final results. Here, ensemble simulation based on multiple reanalyses allows exploring the relative contribution that reanalysis quality has on the overall uncertainty of final results obtained for variables related to land-surface processes or phenomena. Additionally, the average of ensemble members has been shown to improve predictive accuracy relative to individual simulations (e.g., Tebaldi and Knutti, 2007; McGuire et al., 2016). One of the most widely-known ensemble simulation examples is the Coupled Model Intercomparison

Project (CMIP), which aims to contribute to the understanding of past, present, and future climate changes, in a multi-model context.

## 3  GlobSim

### 3.1  Structure and approach

GlobSim is a Python (version 3.7) software package designed to download and process important global atmospheric reanalyses

and to derive time series with consistent variables, units and time intervals for specific locations (Figure 1). It comprises three parts. (i) Downloading for retrieving original data. (ii) Interpolation of original, gridded variables to point locations as time-series. (iii) Scaling of site-level time-series to common units and temporal resolution, and possibly, the application of additional, empirical downscaling function. GlobSim is designed to be controlled via simple parameter files containing keyword-value pairs (e.g. download area, date range, output locations, variables). As downscaling methods can be easily added, it provides a

basis for broader application and development of existing methods Fiddes and Gruber (2012, 2014); Sen Gupta and Tarboton (2016); Cao et al. (2017).

### 3.2  Reanalyses

The four reanalyses currently implemented in GlobSim (Table 1) are ERAI, ERA5, the Japanese 55-year Reanalysis JRA-55 and the Modern-Era Retrospective analysis for Research and Applications (version 2) MERRA-2. Earlier reanalyses, such as

CFSR, ERA-40, JRA-25, and MERRA, were not implemented in GlobSim because they have either been superseded by newer products or have ended. During the writing of this contribution, the discontinuation of ERAI in 2019 has been announced.

ERAI has 60 levels in the vertical dimension with the highest at 1 mb, and the data is interpolated to 37 pressure levels (Dee et al., 2011). A reduced Gaussian grid with approximately uniform 79 km (T255) spacing for surface and other grid-point fields is used. ERAI covers the period from 1 January 1979 to 2019. The data contains analyses (at 00:00, 06:00, 12:00 and 18:00

UTC) for surface and pressure levels as well as forecasts of instantaneous (e.g. 2-m surface air temperature, air temperature,





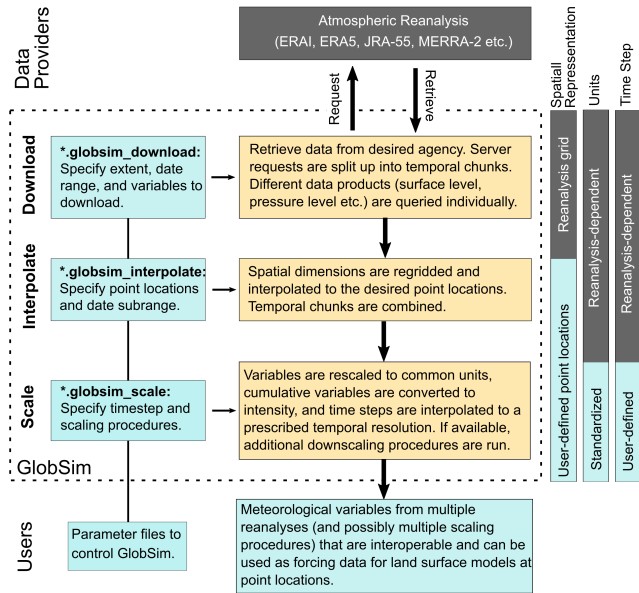

**Figure 1.** GlobSim schematic: Reanalysis data is processed in three main steps, based on user-specified parameter files. The colour of each box identifies user input and output (light blue), GlobSim processing (orange) and raw reanalysis data (grey).

**Table 1.** Characteristics of the four reanalyses currently supported by GlobSim.

| Reanalysis | ERAI | ERA5 | JRA-55 | MERRA-2 |
|---|---|---|---|---|
| Temporal coverage | 1979–2019 | 1979–present | 1958–present | 1980–present |
| Horizontal resolution (latitude × longitude) | 0.75°×0.75° ~79 km | 0.25°×0.25° ~31 km | 0.56°×0.56° ~60 km | 0.5°× 0.625° ~50 km×65 km |
| Vertical levels | 37 | 37 | 27/37 | 42 |
| Temporal resolution | SA: 6-hourly SF: 3/6–hourly PL: 6–hourly | 1-hourly | 6-hourly | SA: 1-hourly SF: 1-hourly PL: 6-hourly |
| Assimilation method | 4D-VAR | 4D-VAR | 4D-VAR | 3D-VAR |
| Sponsoring agencies | ECMWF | ECMWF | JMA | NASA |
| Reference | Dee et al. (2011) | Hersbach and Dee (2016) | Kobayashi et al. (2015) | Gelaro et al. (2017) |

Complete ERA5 data from 1950 to present is expected to be available in late 2019. ERAI = ERA-Interim, SA = surface analysis, SF = surface forecast, and PL = pressure level. ECMWF = European Center for Medium-range Weather Forecast, JMA = Japan Meteorological Agency, and NASA = National Aeronautics and Space Administration. In ERAI, the temporal resolution of surface forecasts is 6-hourly for instantaneous variables (e.g. temperature and air pressure) and 3-hourly for the accumulated variables (e.g. precipitation, short- and long-wave radiation.)





and relative humidity) and accumulated (e.g. total precipitation and radiation components; from 00:00 and 12:00, in 3, 6, 9, and 12-hour steps) variables.

ERA5 is the fifth-generation atmospheric reanalysis produced by ECMWF to replace ERAI. Data is currently available from 1979 onward and later in 2019 is expected to be available starting in 1950. ERA5 is produced with a horizontal resolution of

31 km, a temporal resolution of 1 hour, and 137 vertical model levels, although the output pressure levels are identical to ERAI (Hersbach and Dee, 2016). ERA5 assimilates improved input data that better reflects observed changes in climate forcing, as well as many new or reprocessed observations that were not available during the production of ERAI. It also provides an estimate of uncertainty based on a ten-member ensemble with a temporal resolution of 3 hours and spatial resolution of 62 km (Albergel et al., 2018).

JRA-55 is produced using a 4-dimensional data assimilation system that uses many types of satellite data (Kobayashi et al., 2015). It is the second Japanese atmospheric reanalysis and covers the period from 1958 to near real-time. JRA-55 has a spatial resolution of 1.25° for assimilation (TL319) and 37 vertical pressure levels (1000–1 mb) for most variables in upper air, except dew-point depression, specific humidity, relative humidity, cloud cover, cloud water, cloud liquid water, and cloud ice, which are produced for 27 levels from 1000 to 100 mb, only. The temporal resolution is 6 hours for all levels and data (Kobayashi

et al., 2015).

MERRA-2 replaces the original MERRA reanalysis (Gelaro et al., 2017) and uses the Goddard Earth Observing System-5 (GEOS-5) general circulation model (GCM) (Molod et al., 2015). It uses the cubed sphere grid of Putman and Lin (2007) and has a spatial resolution of 0.5°×0.625° (latitude × longtitude, ∼50 km). MERRA-2 has 42 consistent pressure levels from the surface up to 0.1 mb (Gelaro et al., 2017). It has a temporal resolution of 6 hours for the pressure level data and 1 hour for

surface and forecast analysis.

## 3.3 Operation

### 3.3.1 Download

GlobSim downloads reanalysis data from each sponsoring agency (Table 1). Downloads are mediated through application program interfaces (APIs) which allow programmatic access to data servers through Python. Server requests are split into

temporal chunks in order to efficiently download large datasets. Different types of data (surface analysis, surface forecasts, and pressure level analysis) are queried individually. The outputs are stored as netCDF4 files for each temporal chunk and reanalysis field, retaining the structure and conventions used by the sponsoring agencies.

### 3.3.2 Spatial interpolation

GlobSim spatially interpolates gridded variables to the latitude and longitude of point locations using bilinear interpolation

through the Python interface (ESMPy) of the Earth System Modeling Framework (ESMF) toolkit (O'Kuinghttons et al., 2016). Pressure-level variables are also interpolated spatially on each pressure level and in a subsequent step, vertical interpolation is performed at that location. First, geopotential height is normalized to obtain pressure-level elevation ($E$) [m] for each time step





$$E = \frac{\phi}{g_0},\tag{1}$$

where $\phi$ is the geopotential height [m$^2$ s$^{-2}$] and $g_0$ is the acceleration due to gravity of 9.807 m s$^{-2}$. Then, pressure-level variables are vertically interpolated to the desired elevation. Geopotential height and other pressure-level variables are linearly

extrapolated to locations where the pressure is greater than that of the lowest level with reanalysis data based on the values of the two lowest available pressure levels (Yessad, 2018).

### 3.3.3 Scaling

We use the term scaling to describe the conversion of variables and units from their reanalysis-specific origins to a common standard as well as the possible application of additional temporal interpolation and downscaling procedures to produce data

at fine spatial and temporal scales. For example, 3-hourly short-wave radiation from reanalyses can be interpolated to hourly resolution and used at mountain locations. When additionally, fine-scale terrain-sun geometry and horizon shading are taken into account the resulting data is likely to be more accurate (Fiddes and Gruber, 2014).

The output variables of GlobSim are named following CF conventions (v1.6) with standard_name attributes that are consistent with UDUnits. The units of meteorological variables are first converted to the standard ones and then interpolated to achieve

the required temporal resolution. Some of the forecast variables are not directly obtained from the reanalyses but instead are derived based on calculations involving other available variables (e.g. wind speed may be calculated from its northward and eastward components).

All analysed fields in ERAI (e.g., surface analysis and pressure levels) and many forecast fields (e.g. temperature) are instantaneous. Some forecast variables (e.g., precipitation amount, surface downwelling longwave flux in air) however, are

provided as accumulations in the reanalyses (Figure 2). This means that each value is described as a change relative to another time in the forecast cycle instead of as an instantaneous flux. In contrast to the other reanalyses, the forecasts in ERAI are accumulated from the beginning of the respective forecast cycle (i.e. from 00:00 or 12:00, the solid blue lines in Figure 2) rather than from the last step of the previous forecast cycle.

In the scaling procedure, ERAI forecasts are first disaggregated to total amounts starting from the end of each previous step

(the dashed blue lines in Figure 2) for respective forecast cycle (00:00 and 12:00), and can be expressed as

$$S_s^d = \begin{cases} I_s, & s = 3 \\ I_s - I_{s-1}, & s = 6, 9, 12, \end{cases}\tag{2}$$

where $I$ denotes values spatially interpolated from the reanalysis grid (see Section 3.3.2) and, $S$ is the scaled results for that point location. The subscript $s$ denotes the step and $d$ the disaggregated value. The accumulated variables in forecasts are then converted to averages at the prescribed time resolution by dividing the length of the time step

$$S = \frac{S_s^d}{t_n},\tag{3}$$



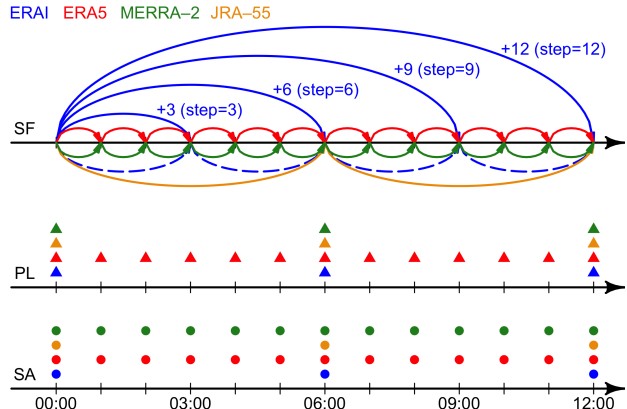

**Figure 2.** The representation of surface analysis (SA), surface forecasts (SF), and pressure levels (PL) data in the four reanalyses supported by GlobSim. The circles and triangles represent instantaneous variables which are reported at specific temporal resolutions. The lines represent accumulated variables in the forecasts which are reported as the integral of a flux over the time interval spanned by each arrow. Because the surface forecasts in ERAI are accumulated from the beginning of the respective forecast cycle (the solid blue lines) rather than from the end of the previous step, they must be disaggregated (the dash blue lines) using Eq. 2 to prevent overlap in the integration periods.

where the temporal scale factor ($t_n$) could be given as

$$t_n = \frac{t_{res}}{t_{out}},$$ (4)

where $t_{res}$ and $t_{out}$ is the raw temporal resolution of reanalysis and required temporal resolution of outputs with the unit of hour [h]. For ERAI, the disaggregated downwelling shortwave flux in air was found to be slightly negative ($> -0.05$ W m$^{-2}$) at 5 some time steps. These values were interpreted to be artifacts and set to zero.

The surface downwelling longwave flux in air ($LW_d$) is not directly available from MERRA-2, and is instead calculated as the sum of the longwave flux emitted from surface ($LW_e$) and the surface net downward longwave flux ($LW_n$),

$$LW_d = LW_e + LW_n.$$ (5)

In ERAI, relative humidity (RH) [%] is derived by following Lawrence (2005)

10 $$RH = 100 - 5 \times (T_a - T_d)$$ (6)

where $T_a$ [K] is the near-surface air temperature and $T_d$ [K] is the dewpoint temperature. Wind speed and direction are calculated from the eastward (U) and northward (V) components

$$W_s = \sqrt{U^2 + V^2}$$ (7)

and

15 $$W_d = atan2(V, U) \times \frac{180}{\pi} + 180$$ (8)





where $W_s$ is the wind speed in m s$^{-1}$, $W_d$ is wind direction in degrees and $atan2$ is the two-argument arctangent used to compute an unambiguous angle when converting from Cartesian to polar coordinates.

## 4 Demonstrator application

Reanalysis products are carefully designed and tested before release. In addition, many studies have evaluated their performance by inter-comparison, by comparison with observations (e.g., Jiang et al., 2015) and by applying them to model simulation (e.g. Albergel et al., 2018; Beck et al., 2019). For this reason, we focus on the application of GlobSim for demonstration rather than the direct testing of reanalysis variables or their interpolated products. An inter-comparison of GlobSim-derived meteorological variables for the test area (Fig. 3) helps to appreciate differences or detect blunders in conversion. Below, we use GlobSim to drive ground-surface temperature (GST) simulation using a permafrost/land-surface model for a remote location in Northern Canada underlain by permafrost. We then analyze ensemble results and their deviance from observations.

### 4.1 Study area

The research area is centered at 64°42' N, 110°36' W, near the north shore of Lac de Gras in the Northwest Territories, Canada (Figure 4). The area is located within the zone of continuous permafrost, the mean annual air temperature (MAAT) is -9.0 °C and the total annual precipitation is 284 mm during 09/1998–08/2007 (Ekati A, Environment Canada, 2019), with about 50% occurring as snow (Jones et al., 2003). Typically, snow cover lasts for about seven months and because of strong winds (Hu et al., 2003), snow depth shows strong spatial variability; it is shallowest in much of the higher-elevation, convex terrain (e.g. tops of eskers) and deepest in low-lying areas with taller shrub vegetation or in the lee of larger terrain features (Holubec et al., 2003).

The area has undulating to moderately rugged topography dominated by glacial features (Kerr et al., 1997; Dredge et al., 1999), mostly on the order of 10–20 m in relief (Dredge et al., 1999) and composed of glacial till (Haiblen et al., 2018). Till deposits are described according to their thicknesses, as veneers (< 2 m), blankets (2–10 m) or hummocky (5–30 m). Eskers are the prevalent glaciofluvial deposits, reaching heights of 35 m and occasionally containing massive ice on the order of 2–5 m thick (Haiblen et al., 2018; Wolfe et al., 1997). The poorly drained low-lying areas have peat deposits and are generally associated with ice-wedge polygons. The region is within the Southern Arctic Ecozone (Wiken et al., 1993) described by continuous shrub tundra (Wiken et al., 1996). The most common shrubs are dwarf birch (*Betula pumila*) and Labrador tea (*Ledum decubens*). Uplands are well-drained with lichen and mosses whereas wetlands are typically colonized with sedges and mosses.

### 4.2 Observations and quality control

GST was measured at 156 locations in order to capture the fine-scale spatial variability and to test the performance of GlobSim in supporting GST simulation beneath different terrain classes. Study plots measuring 15 m × 15 m were established that reflect the different terrain types in in the area (Gruber et al., 2018). Each plot was instrumented with three to four temperature



**Figure 3.** Inter-comparisons of monthly variables derived from GlobSim which are used in GEOtop during the period 09/2015–08/2017. $SW_d$ and $LW_d$ is the surface downwelling shortwave flux in air and downwelling longwave flux in air, respectively.



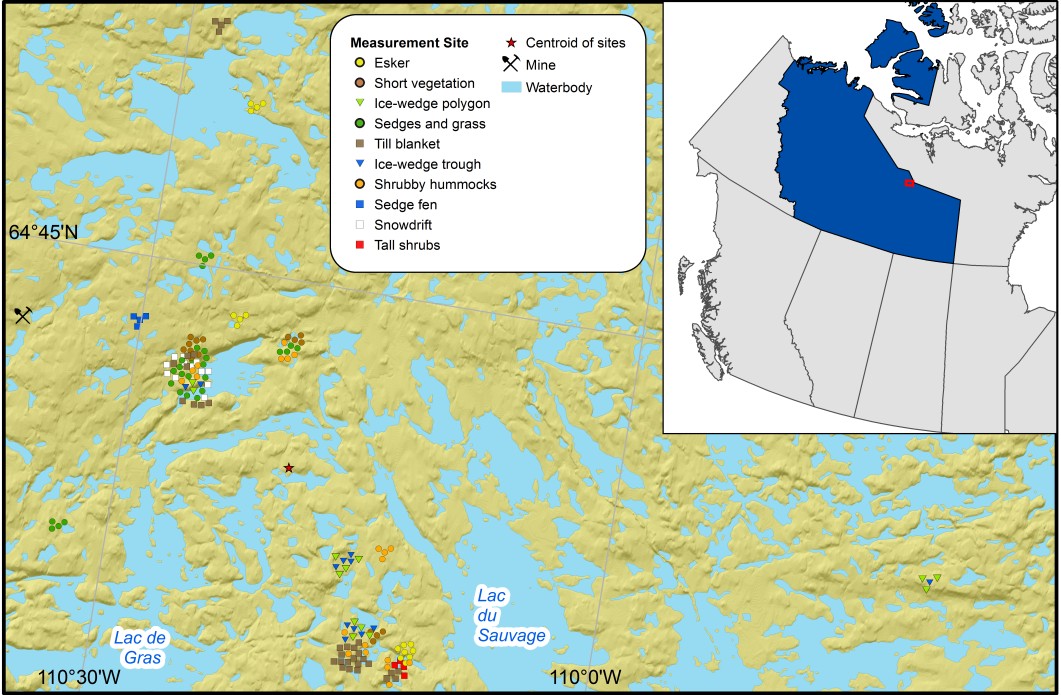

**Figure 4.** Study area and approximate location of measurement sites. Many point locations overlap at the scale of the map, so the positions of their markers have been dispersed to better illustrate the number and type of sites. The centroid of all measurement sites (red star) was used as the location for which GlobSim reanalysis data were derived. The top right inset map shows the location of the study area (red square) in western Canada. The hillshade basemap is derived from the Canadian Digital Elevation Model (CDEM) provided by the Natural Resources Canada Federal Geospatial Platform. Waterbody outlines were obtained from the ArcGIS online data library.

loggers approximately 0.1 m below the ground surface. Site characteristics including surficial geology, topography and snow deposition tendency were recorded for each study plot, and sub-plot characteristics were collected for each logger at the 1 m × 1 m scale including drainage tendency, vegetation height and leaf area index (LAI). Surface air temperature is measured at six locations, equally distributed between topographic high and low points. Sensors are mounted in passively-ventilated radiation

5   shields (Young, model 41003) 2–3 m above the ground surface.

This study uses temperature measurements over two years (09/2015–08/2017) and annual mean values described here refer to measurements beginning in September and ending in August of the following year. GST data loggers (GeoPrecision, model M-Log5W-SIMPLE) and surface air temperature loggers (GeoPrecision, model M-Log5W with a Rotronic Hygroclip sensor) have a resolution of 0.01 °C and an accuracy of ± 0.1 °C. All measurement have intervals of 20 minutes.





## 4.3 Distinguishing terrain types

The variability of GST regimes near Lac de Gras is controlled by many factors, the most significant of which are surficial geology, vegetation type and height and topography (cf. Hu et al., 2003). The 156 sites are hence grouped into ten classes based on surface and subsurface characteristics as described at a scale of 15 m × 15 m (Fig. 5). This classification is subjective and was developed to identify common terrain types that could be easily recognized in the field and that explain a significant part of the observed spatial variation of GST. Surface offset is defined is used here to quantify local ground temperature variations (Smith and Riseborough, 2002). It is defined as $SO = MAGST - MAAT$, where MAGST is the mean annual ground-surface temperature and MAAT is the mean annual air temperature.

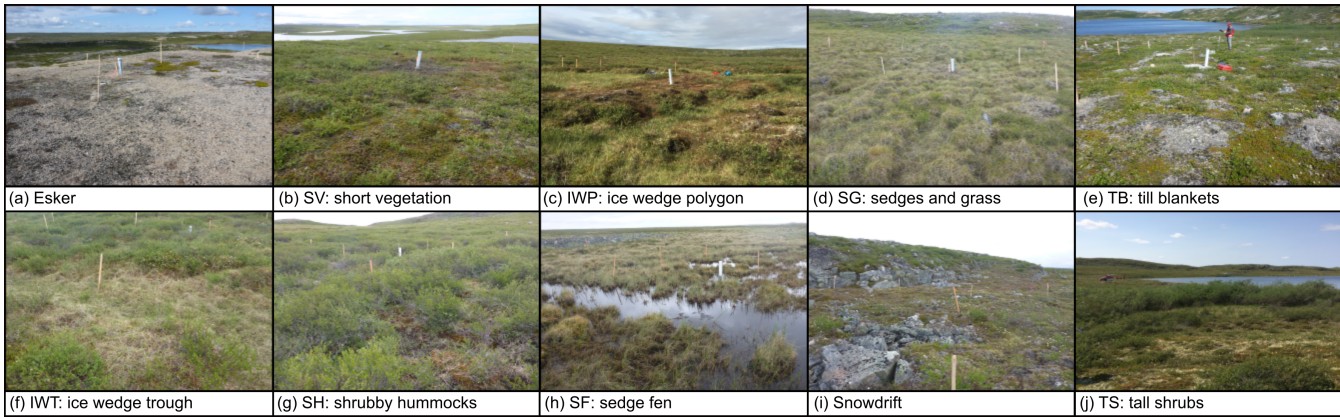

**Figure 5.** The ten terrain types used to partition the landscape and observations.

## 4.4 Process-based numerical model

GEOtop (version 2.0), a physically-based numerical model, is used for this demonstrator application because it describes the complex abiotic processes in permafrost environments well (e.g. Endrizzi and Marsh, 2010; Gubler et al., 2013; Endrizzi et al., 2014; Pan et al., 2016). It represents the heat and water transfer in soil as well as the energy transfer between the soil and the atmosphere. Additionally, it solves equations describing the interactions of water and energy during soil freezing and thawing (Dall'Amico et al., 2011; Endrizzi and Gruber, 2012) and simulates the water and energy transfer in the snow cover. GEOtop, driven by ERA-Interim forcing data, was found to be suitable for reproducing ground temperature observations (Mugford et al., 2012; Fiddes et al., 2015).

## 4.5 Model settings and parameters

GEOtop was forced by the four reanalyses obtained via GlobSim, providing hourly data for precipitation, wind velocity, wind direction, short- and long-wave radiation, relative humidity, and near-surface air temperature. Given the gentle topography and small test area, the time-series were derived for the geographic centre of measurement sites, only (Figure 4). Model parameters





**Table 2.** GEOtop soil parameters used to simulate different stratigraphic units.

| Stratigraphy | Glaciofluvial | Sandy till | Silty till | Peat | Ice-wedge | Bedrock |
|---|---|---|---|---|---|---|
| Saturated water content [%] | 37.5 | 50 | 39.7 | 85 | 95 | 5 |
| Residual water content [%] | 4.9 | 4.9 | 5.3 | 20 | 0 | 1 |
| Van Genuchten $\alpha$ [m$^{-1}$] | 2.6 | 2.1 | 0.8 | 30 | 30 | 1 |
| Van Genuchten $n$ [-] | 1.79 | 1.57 | 1.51 | 1.80 | 1.80 | 1.20 |
| Thermal capacity [$10^6$ J m$^{-3}$ K$^{-1}$] | 2 | 2 | 2 | 1.8 | 1.8 | 2 |
| Thermal conductivity [W m$^{-1}$ K$^{-1}$] | 4 | 2 | 2 | 0.22 | 0.22 | 2 |

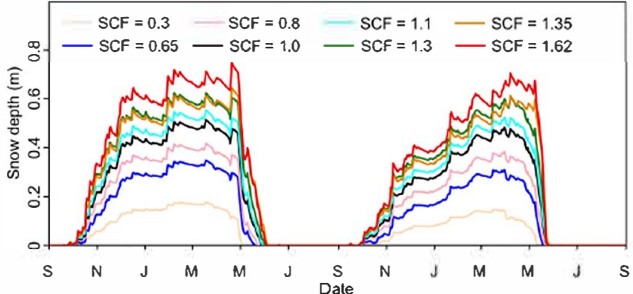

**Figure 6.** Simulated ensemble mean daily snow depth using different snow correction factors (SCFs) to reproduce the snow melt-out date for terrain types during the period 09/2015-08/2017. SV = short vegetation, TB = till blankets, IWP = ice-wedge polygons, SG = sedges and grass, IWT = ice-wedge troughs, SH = shrubby hummocks, SF = sedge fen and TS = tall shrubs. Max is the maximum snow depth during the measured period in meter.

and initial conditions were specified for each terrain type based on field observations. For example, the uppermost soil type was derived from drill logs and soil pit measurements (Table 2). Vegetation height was measured within each 1 m × 1 m sub-plot around the datalogger (Table 3). Some parameters, which are challenging to measure directly, were then subsequently refined, within the range of plausible values (cf. Gubler et al., 2013), through experimentation with the model. The snow correction

5 factor, which scales the simulated amount of snow via precipitation amount, is used to capture the differences among the ten terrain classes. It was determined by fitting the simulated melt-out date of the snow (Schmid et al., 2012) to observations (Fig. 6). Snow blowing is parameterized as wind compaction in 1D (Pomeroy et al., 1993) for all the terrain types except the tall shrubs site.

Soil parameters were estimated based on soil textural data collected at each plot (Subedi, 2016). More specifically, the soil

10 particle thermophysical properties (e.g. thermal conductivity and heat capacity) were estimated based on typical values for common material types in each unit. The soil freezing characteristic curve parameters (van Genuchten $\alpha$, van Genuchten $n$, saturated and residual water content) for rock and organic material were taken from Gubler (2013). For mineral soil, values were obtained by averaging the soil textural data within each class and applying the ROSETTA model v3.0 (Zhang and Schaap,


**Table 3.** GEOtop parameters for different terrain types.

| Terrain | Esker | SV | IWP | SG | TB | IWT | SH | SF | Snowdrift | TS |
|---|---|---|---|---|---|---|---|---|---|---|
| **Soil parameters** | | | | | | | | | | |
| Stratigraphy | I | II | III | III | II | IV | V | III | II | II |
| Soil water content [%] | 12 | 45 | 85 | 77.5 | 40 | 95 | 36 | 85 | 45 | 45 |
| | 15 | 50 | 39.7 | 39.7 | 50 | 39.7 | 39.7 | 39.7 | 50 | 50 |
| | 0.05 | 0.05 | 0.05 | 0.05 | 0.05 | 0.05 | 0.05 | 0.05 | 0.05 | 0.05 |
| Simulation depth [m] | | | | | 12 | | | | | |
| Initial ground temperature [°C] | | | | | -2 | | | | | |
| **Snow parameters** | | | | | | | | | | |
| Snow correction factor [-] | 0.3 | 0.65 | 1 | 1.1 | 0.8 | 1.1 | 1.3 | 1.3 | 1.62 | 1.35 |
| Wind compaction [-] | yes | yes | yes | yes | yes | yes | yes | yes | yes | no |
| **Vegetation parameters** | | | | | | | | | | |
| Vegetation height [m] | 0.15 | 0.20 | 0.20 | 0.15 | 0.30 | 0.20 | 0.30 | 0.35 | 0.15 | 2.00 |
| LSAI [$m^2 m^{-2}$] | 1.5 | 2.3 | 4.1 | 3.7 | 2.7 | 2.1 | 3.5 | 2.1 | 3.0 | 3.8 |
| Canopy fraction [–] | 0.4 | 1.0 | 0.8 | 0.8 | 0.7 | 0.8 | 0.8 | 0.6 | 0.8 | 1.0 |
| DecayCoeffCanopy [–] | 2.5 | 2.7 | 2.7 | 2.5 | 3 | 2.7 | 2.7 | 3.2 | 2.5 | 3.5 |
| Vegetation root depth [m] | 0.10 | 0.10 | 0.15 | 0.10 | 0.20 | 0.15 | 0.20 | 0.10 | 0.10 | 0.7 |
| Vegetation reflectivity in the visible [–] | | | | | 0.11 | | | | | |
| Vegetation transmissivity in the visible [–] | | | | | 0.55 | | | | | |
| Vegetation reflectivity in the near infrared [–] | 0.45 | 0.50 | 0.45 | 0.40 | 0.45 | 0.40 | 0.50 | 0.40 | 0.45 | 0.45 |
| Vegetation transmissivity in the near infrared [–] | 0.25 | 0.25 | 0.30 | 0.34 | 0.25 | 0.30 | 0.25 | 0.34 | 0.25 | 0.25 |
| Surface density of canopy [$kg\,m^{-2}$] | 0.4 | 1.6 | 3.6 | 3.5 | 1.7 | 1.6 | 4.1 | 2.8 | 3.5 | 4.0 |
| VegSnowBurying [–] | 2.0 | 2.0 | 1.5 | 1.0 | 2.0 | 1.5 | 2.0 | 1.0 | 2.0 | 2 |

SV = short vegetation, IWP = ice-wedge polygons, SG = sedeges and grass, TB = till blankets, IWT = ice-wedge troughs, SH = shrubby hummocks, SF = sedge fen, TS = tall shrubs. The soil water content is specified for each soil layers present in Figure 7. DecayCoeffCanopy = Decay coefficient of the eddy diffusivity profile in the canopy, and VegSnowBurying = Coefficient of the exponential snow burying of vegetation. The different stratigraphic profiles are listed in Figure 7.

2017). For the ice-wedge trough, the uppermost layer was assigned a the water/ice content of 95% to simulate the ice-wedge itself. Leaf and stem area index (LSAI) and surface density of canopy were estimated based on measured LAI and moss thickness. Canopy fraction was determined based on plot photos (Gruber et al., 2018). Although the same vegetation reflectivity (0.11) and transmissivity (0.15) was used for each terrain type, overall reflectivity and transmissivity are in fact dependent on

5    vegetation and subsurface conditions.





A lower boundary condition of zero heat flux was used for these shallow simulation. Although borehole analysis revealed a heat flow of 0.046 W m$^{-2}$ based on temperature measurements in two deep boreholes near Lac de Gras (Mareschal and Jaupart, 2004), it is not appropriate to apply these measurements to the shallow profile simulations because of the transient changes in the temperature profile near the surface during recent decades. We use 30 soil layers with a total depth of 12 m (Figure 7) and an intial soil temperature of -2 °C. Reanalyses data for 07/2000–06/2010 was used to spin up the model by running it ten times (100 years) before simulations are conducted from 07/2010 onward.

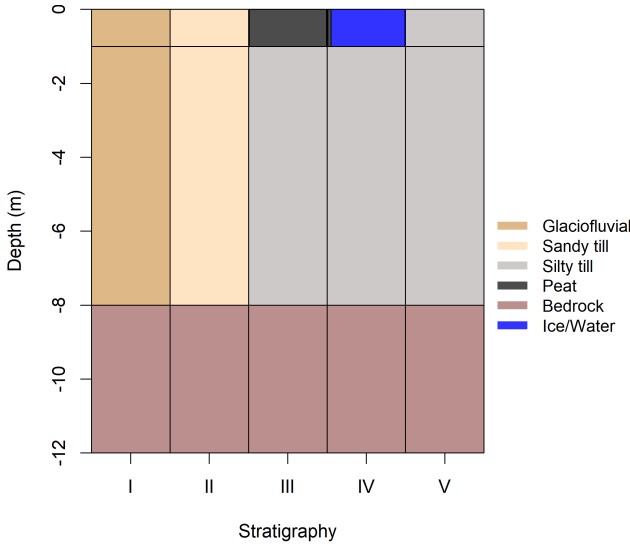

**Figure 7.** Soil profiles for the different terrain types used to partition surface temperature observations. Parameters for each of the subsurface materials are provided in Table 2. The stratigraphic unit (I–V) associated with each of the terrain types is listed in Table 3.

Reanalysis produces multi-decadal meteorological variables (Table 1), and this makes simulating long-term changes of land-surface processes possible. To demonstrate the utility of GlobSim for supporting long-term simulation, we conducted an additional ground temperature simulation from 1980 to 2017 for a single terrain type and the model was spun up by repeating the reanalysis of 01/1980–12/1984 100 times (500 years).

### 4.6 Comparison of observation and simulation.

To compare simulations with observations, the mean bias (BIAS) and root mean squared error (RMSE) were computed for each time series as

$$BIAS = \frac{1}{N} \sum_{t=1}^{N} (T_{mod} - T_{obs}) \text{ and} \qquad (9)$$

$$RMSE = \sqrt{\frac{\sum_{t=1}^{N} (T_{mod} - T_{obs})^2}{N}}, \qquad (10)$$

where $T_{mod}$ is the modeled temperature, $T_{obs}$ is the observed temperature, and $N$ is the total number of measurements.





## 5 Results

### 5.1 Observed temperature

During the two years measured, the MAGST (SO) at individual sites span a large range of 11.43 °C, from -8.03 (0.46) °C on

an esker with low vegetation, soil moisture, and snow cover (Figure 6), to 3.40 (11.89)°C in the tall shrubs, which retain snow

5 (Figure 5, Table 4). Daily GST shows seasonal patterns that are similar for each terrain type, MAGST ranges within terrain

types are reduced to 1.19–6.50°C and mean surface offsets differ clearly.

**Table 4.** Summary of temperatures in the ten terrain types.

| Terrain | | All | Esker | SV | IWP | SG | TB | IWT | SH | SF | Snowdrift | TS |
|---|---|---|---|---|---|---|---|---|---|---|---|---|
| Number of sites | | 156 | 15 | 16 | 13 | 28 | 32 | 12 | 20 | 4 | 12 | 4 |
| MAAT (°C) | | | | | | -8.49 (2016); -8.15 (2017) | | | | | | |
| MAGST (°C) | max | 3.40 | -3.85 | -2.10 | -0.33 | 0.76 | 1.41 | 1.31 | 2.20 | 1.13 | 2.04 | 3.40 |
| | min | -8.03 | -8.03 | -4.43 | -4.48 | -4.98 | -5.06 | -2.58 | -1.70 | -0.06 | -1.84 | -1.50 |
| | mean | -1.77 | -5.43 | -3.46 | -2.94 | -2.37 | -2.16 | -0.65 | 0.48 | 0.50 | 0.76 | 1.37 |
| | sd | 2.21 | 1.01 | 0.73 | 1.08 | 1.34 | 1.72 | 1.18 | 1.01 | 0.47 | 0.93 | 2.09 |
| SO (°C) | max | 11.89 | 4.64 | 6.39 | 8.16 | 9.25 | 9.90 | 9.80 | 10.69 | 9.62 | 10.53 | 11.89 |
| | min | 0.46 | 0.46 | 3.72 | 3.67 | 3.51 | 3.09 | 5.57 | 6.45 | 8.09 | 6.31 | 6.65 |
| | mean | 6.55 | 2.89 | 4.85 | 5.38 | 5.95 | 6.16 | 7.66 | 8.80 | 8.82 | 9.07 | 9.69 |
| | sd | 2.24 | 1.02 | 0.87 | 1.17 | 1.39 | 1.75 | 1.29 | 1.12 | 0.64 | 1.03 | 2.13 |

SV = short vegetation, IWP = ice-wedge polygons, SG = sedeges and grass, TB = till blankets, IWT = ice-wedge troughs, SH = shrubby hummocks, SF = sedge

fen and TS = tall shrubs. MAAT = mean annual air temperature, MAGST = mean annual ground surface temperature, and SO = surface offset.

### 5.2 Simulated temperature

The surface air temperature is best approximated by ERA5 which has the lowest daily RMSE of 1.94°C and bias of 0.15°C,

while MERRA-2 was the worst with an RMSE of 3.65°C and a bias of -1.62°C (Table 5). During 2015–2017, the mean MAAT

10 derived from the four reanalyses was -8.11±1.25°C which is in good agreement with the observed MAAT of -8.32°C. The

ensemble mean, which had a daily RMSE of 1.87°C outperformed all individual reanalyses, during the measured period and

reduced RMSE by 0.07–1.78°C (Figure 8).

Figures 8c–l compare the daily ensemble means of simulated GST to observation means for the ten terrain types. The

performance varies significantly among terrain types and reanalyses, with the RMSE ranging from 1.09°C to 3.00°C (Table 5).

15 Even within the same terrain class, the mean RMSE difference for the four reanalyses was 0.60±0.22°C and was up to 0.89°C

for the sedge fen terrain type. The overall RMSE for the four reanalyses was 1.96°C for daily means and 0.92°C for annual

means.





Most (81%) of the simulated daily GST are within the observation range, although the spread in simulated values is generally smaller than in observations (Figure 8). The spread of model results is generally greatest sometime between January and May of each year, with the exact timing depending on the terrain type. The variability of observations also depends on the terrain type, with the lowest variability at sedge fen sites and the greatest at till blanket sites. The ensemble mean shows good agreement
5 with the observation mean with RMSE ranging from 1.02°C to 2.45°C depending on the terrain type.

The GST ensemble mean usually performed better compared to the individual ensemble members based on single reanalyses, as was the case with surface air temperature. For six of ten terrain types, the simulated GST ensemble mean achieved better results based on the RMSE. Moreover, the ensemble mean had the smallest RMSE for the daily GST when averaged over all sites. Consequently, the RMSE was reduced by 0.01–0.31°C for daily GST, and by 0.00–1.01°C for SO as a whole (Figure 9).
10 Overall, simulations forced by MERRA-2 achieved the best performance for MAGST with the BIAS and RMSE of -0.03 and 0.75°C, respectively. Compared to other individual reanalyses, the simulations forced by ERA5 often yield the second best RMSE (8/14) if not the best (2/14) (Table 5).

**Table 5.** Deviance of simulations from the mean of observations within each terrain type. The smallest BIAS and RMSE values for each row are indicated in bold font.

| Source | Classes | ERAI BIAS | ERAI RMSE | ERA5 BIAS | ERA5 RMSE | JRA-55 BIAS | JRA-55 RMSE | MERRA-2 BIAS | MERRA-2 RMSE | Ensemble BIAS | Ensemble RMSE |
|---|---|---|---|---|---|---|---|---|---|---|---|
| SAT | | 1.18 | 2.19 | **0.15** | 1.94 | 1.11 | 2.30 | -1.62 | 3.65 | 0.20 | **1.87** |
| | Esker | 1.57 | 2.78 | 0.95 | **2.27** | 0.71 | 2.49 | **-0.03** | 3.00 | 0.80 | 2.45 |
| | SV | 0.91 | 2.75 | 0.59 | 2.41 | 1.29 | 2.45 | **0.00** | 2.72 | 0.70 | **2.39** |
| | IWP | 0.80 | 2.19 | 1.03 | 1.92 | 1.50 | 2.22 | **0.58** | 1.95 | 0.98 | **1.87** |
| | SG | **0.82** | 1.77 | 1.26 | 1.76 | 1.60 | 2.31 | **0.82** | 1.80 | 1.12 | **1.72** |
| | TB | 0.98 | 2.07 | 1.04 | **1.72** | 1.49 | 2.42 | **0.68** | 2.03 | 1.05 | 1.90 |
| GST | IWT | **-0.11** | 1.34 | 0.50 | 1.42 | 1.08 | 2.22 | 0.31 | 1.55 | 0.44 | **1.27** |
| | SH | -0.24 | 2.06 | **-0.07** | 1.69 | 0.22 | 1.66 | -0.73 | 1.93 | -0.21 | **1.61** |
| | SF | -0.41 | **1.15** | 0.02 | 1.48 | 0.51 | 2.04 | -0.11 | 1.70 | **0.00** | 1.36 |
| | Snowdrift | -0.32 | 1.52 | -0.13 | 1.09 | 0.51 | 1.18 | -0.33 | 1.57 | **-0.07** | **1.02** |
| | TS | -1.48 | 2.06 | -1.29 | 1.81 | **-0.50** | **1.41** | -1.53 | 2.18 | -1.20 | 1.66 |
| | Overall | 0.25 | 2.04 | 0.39 | 1.79 | 0.84 | 2.09 | **-0.03** | 2.09 | 0.36 | **1.78** |
| MAGST | | 0.39 | 0.91 | 0.25 | 0.91 | 0.84 | 1.08 | **-0.03** | **0.75** | 0.36 | 0.82 |
| SO | | 0.24 | 0.80 | -0.93 | 1.27 | -0.27 | **0.74** | 1.59 | 1.75 | **0.16** | **0.74** |

SV = short vegetation, IWP = ice-wedge polygons, SG = sedges and grass, TB = till blankets, IWT = ice-wedge troughs, SH = shrubby hummocks, SF = sedge fen, and TS = tall shrubs. SAT = daily surface air temperature, GST = daily ground-surface temperature, MAGST = mean annual ground surface temperature, and SO = surface offset.





**Figure 8.** Comparison of ensemble (ENS) simulation results with observations (OBS) for (a) daily near-surface air temperature (SAT); (b) daily ground surface temperature (GST) for all the sites and; (c–l) daily GST for individual terrain classes. The date range for all figures is September 2015 to September 2017.

The long-term simulation shows warming trends for MAAT and for MAGT at all depths. The modeled warming rate of the ensemble mean was 0.42 °C per decade for MAAT and 0.42, 0.21, and 0.13 °C per decade for the annual mean ground temperature at depths of 0.1 m, 10 m, and 20 m, respectively (Fig. 10).



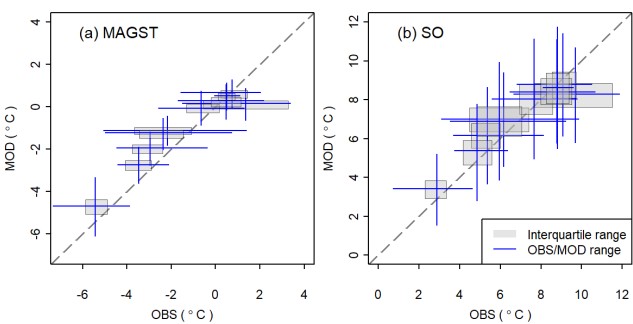

**Figure 9.** Comparison of ensemble results and observations for (a) mean annual ground surface temperature (MAGST) and (b) surface offset (SO) for the ten terrain types.

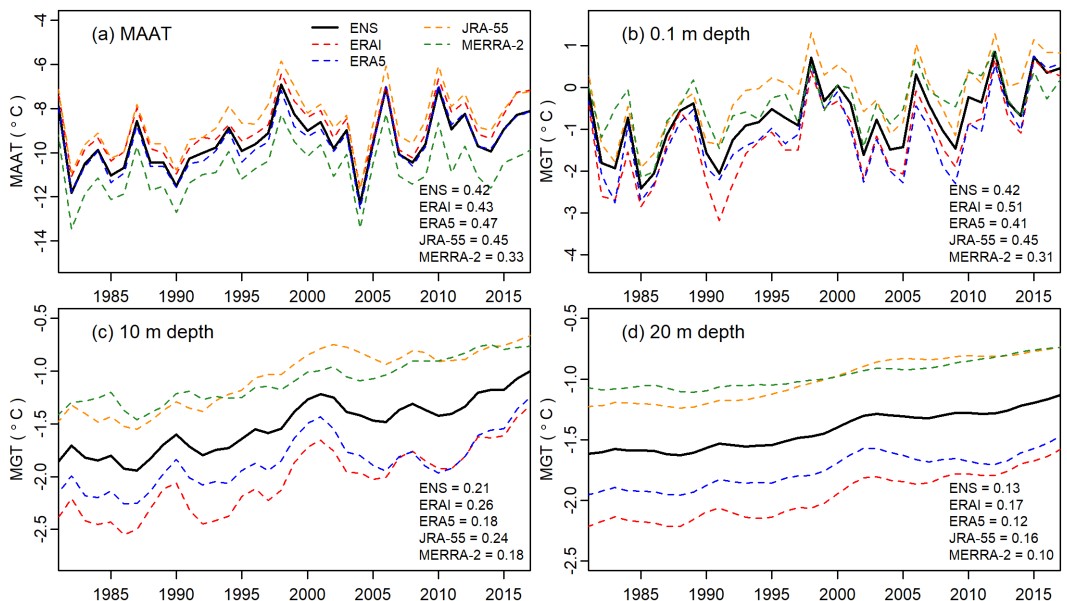

**Figure 10.** Changes of (a) mean annual air temperature (MAAT) derived from reanalyses, modeled annual mean ground (MGT) temperature at the depth of (b) 0.1 m, (c) 10 m, and (d) 20 m for shrubby hummocks (SH) terrain type of between 1980–2017. The numbers in the lower right of each figure correspond to the magnitude of the warming trend for each reanalysis and their ensemble mean with the unit of °C per decade. ENS corresponds to ensemble mean. The trend lines were calculated using a linear regression model, and all the slopes were statistically significant with $p < 0.05$.

## 6 Interpretation

The observed spatial variability of GST, with MAGST difference up to 11.5°C over 30 km, highlights the need for spatially-differentiated simulation in order to represent the different thermal regimes among locations or terrain types as well as their different transient responses to climate forcing. The reduced standard deviation and range of observed MAGST within terrain





types indicates that the classification method for terrain type used here is reasonable. Furthermore, it shows that conceptualizing and simulating GST via terrain types can be expected to explain a significant part of the variation observed.

The variables in GlobSim output are similar between reanalyses at one co-located point, indicating that major blunders are unlikely to exist in the GlobSim code. The data approximate surface air temperature near Lac de Gras well and GST simulation results from GEOtop agree well with observations both in terms of summary statistics (Table 5) and the reproduction of the underlying seasonal ground thermal regime (Fig. 8). While quality requirements and simulation quality will differ with application and geographic area, our demonstrator shows the suitability of reanalyses and GlobSim for driving process-based numerical simulation at the site scale. It shows that the combination of driving meteorological data, site conditions and a suitable model are able to reproduce spatial differences as well as the temporal evolution of ground temperature. The availability of multi-decadal reanalysis time series opens the door for better investigating transient processes and phenomena at and below the land surface. Ground temperature is one of many possible variables to investigate and, in this study, was chosen as an exemplar because it reflects the motivation and expertise of the authors.

The greatest model spread occurs between January and May when the thickness of the insulating snowpack accumulates differences in meteorological variables over the winter period and plays an important role in controlling ground temperature. This increase in model spread is also conspicuously absent in the esker terrain type, for which the effect of snow is minimal. The low variability of observed GST for the sedge fen terrain type may be due in part to the small number of measurement sites in this terrain type (n=4). However, the tall shrubs class has the same number of measurements and exhibits variability similar to other terrain types with a higher number of measurements (e.g. shrubby hummocks, n=20). This suggests that the observed spatial variation of GST in each terrain type is a good first-order indicator of its full range of variability.

The relatively small range of GST in the model ensemble when compared to the observations is likely due to the small number of simulations, in which only differing re-analyses are used but not perturbed physics or differing land-surface models. The large difference of RMSE for simulated ground surface temperature based on the four forcing reanalyses indicates the potential uncertainty caused by forcing datasets. Ensemble means often achieve better performance compared to the individual ensemble members, indicating the potential of this approach to improve simulation results. Both findings underscore the value of using more than one reanalysis for driving simulation studies.

MERRA-2 underestimated surface air temperature near Lac de Gras, while simulation overestimated the ground temperature. As a result, the BIAS of MAGST forced by MERRA-2 is very close to 0 and the RMSE is smallest due to opposing biases canceling out. This is also demonstrated by the highest RMSE of surface offset for MERRA-2.

## 7 Discussion

### 7.1 Observed and simulated temperature

Our results highlighted the magnitude of spatial heterogeneity in ground temperature (see also Smith, 1975; Gubler et al., 2011; Schmid et al., 2012; Morse et al., 2012; Gisnås et al., 2014) over distances that usually are within single grid cells





(30–100 km) of a reanalysis or climate model. When comparing grid-scale simulation results with point observations, this heterogeneity and scale mismatch usually confound model validation.

The ensemble ranges of modelled and observed GST (Fig. 8) reflect two distinct sources of variability. The former stems from differences in the forcing data, and the latter is due to terrain characteristics. However, both ranges inform how well

a simulation would represent a particular type of location within the study area. And, while a direct comparison of the two ranges may not be valid, the observed variability helps contextualize the uncertainty introduced by the forcing data relative to changes in GST over as little as a few metres. From this study, we see that the relative importance of reanalysis uncertainty varies seasonally and spatially.

While we do not aim to evaluate the quality of reanalyses themselves, the calculated air temperature bias indicates how

accurate reanalyses and GlobSim are for our study area. It is therefore worthwhile to contextualize these results with those of studies dedicated to a formal evaluation of the reanalyses. For example, in Greenland (Reeves Eyre and Zeng, 2017) found average monthly SAT biases averaged over all stations of 0.81°C (MERRA-2), 1.76°C (ERAI) and 1.95°C (JRA-55). Wang and Zeng (2012) found a daily average bias for ERAI over 63 stations on the Tibet Plateau of 3.21°C while an analysis of individual stations within North America found 6-hourly ERAI bias values to mostly fall between -1.5°C and 4.5°C but were as

large as 7.5°C (Decker et al., 2012). The SAT bias calculated at our sites ranges from -1.62°C (MERRA-2) to 1.18°C (ERAI) (5). The numbers of this and previous studies remind us that the application of reanalyses for simulating surface phenomena is bound to be imperfect and that the success of application in one area cannot be transferred to other areas uncritically.

## 7.2 Advantages and limitations of GlobSim

Previous work has investigated parameter uncertainty with ensemble simulation (Harp et al., 2016; Gubler et al., 2013) but not

investigated the effect of uncertainty in the forcing data. Other studies have used multiple forcing datasets to drive ensembles at the grid-scale. Jafarov et al. (2012) used a five-member GCM composite product for driving a permafrost model on a 2 km × 2 km grid using mean monthly air temperature and precipitation. Here, the coarse temporal resolution and the GCM-specific realization of weather events preclude some types of detailed investigation with observations. Guo et al. (2017) used three different reanalyses to evaluate the effect of forcing data on permafrost model uncertainty on a 0.5° × 0.5° grid, excluding

fine-scale variation. For simulation at finer scales, dynamic and/or statistical downscaling of GCM and RCM outputs have been used (Salzmann et al., 2007; Marmy et al., 2016). The downscaling and debiasing, however, are often limited to areas with detailed observations.

Although the demonstrator presented here is relatively simple, it addresses a critical gap in permafrost research and likely for other modelling communities as well. It provides a basis upon which to implement improved downscaling methods and

to work towards debiasing GCM results with re-analyses (Cannon, 2016) at the point scale. Specifically, GlobSim helps to: (1) *evaluate models* at locations where observations to compare with model results (ground temperature in this demonstrator) are available, but meteorological observations to drive models are lacking, and (2) *predict environmental phenomena* that are driven by atmospheric conditions (permafrost in this demonstrator) at locations for which no observational data exist. Such predictions could also support the analysis and interpretation of field manipulation experiments or long-term monitoring data.



In keeping with the permafrost example, these experiments could investigate the effects of vegetation change or different snow management practices (O'Neill and Burn, 2017). In remote locations long-term meteorological observations are sparse, which limits the application of models that require detailed inputs. The recent publication of two such datasets illustrates the importance—and the general lack—of complete records which can be used to both force and evaluate models (Boike

et al., 2019, 2017). While the methods contained in GlobSim are simple, it nevertheless provides an important simplification of the application of reanalysis data toward simulation studies outside atmospheric science. As GlobSim outputs use the CF conventions (Hassell et al., 2017) it contributes to the ease of using multiple data sources.

Our results have demonstrated the performance of GlobSim for site-level simulation in one particular field area with gentle topography. Simulation accuracy for other areas and application will differ, especially in mountains and near coasts where

topography and other heterogeneity presents additional challenges for downscaling. Additional scaling rules such as TopoScale (Fiddes and Gruber, 2014), REDCAPP (Cao et al., 2017) or those implemented in MSDH (Sen Gupta and Tarboton, 2016) may be added in the future, making GlobSim more suitable in mountains.

### 7.3   ERA5 ensemble

ERA5 provides uncertainty estimates for all parameters at 3-hour intervals and at a horizontal resolution of 62 km. This is

achieved using an ensemble of 10 members that differ in their assimilated observations, initial conditions and model structure. In other words, ERA5 itself is an ensemble simulation. In GlobSim, only the ensemble mean of ERA5 is currently used. In a next step, efforts will be devoted to compare ensembles of ERA5 and fully incorporate them into GlobSim.

### 8   Conclusions

We describe and test the software GlobSim, which has been designed to support ensemble simulations at a the site level.

Specially, GlobSim is designed to easily retrieve, interpolate, and scale reanalyses in order to produce time series of meteorological variables with common structure, temporal resolution and units. It currently supports four reanalyses: ERAI, ERA5, JRA-55 and MERRA-2. We demonstrate the utility of GlobSim by driving a model of ground-surface temperature in a tundra environment with permafrost, and comparing its output to observations. Our results support three conclusions:

1   GlobSim improves the usability of reanalyses for land-surface simulations by deriving time-series of climate variables

25     in uniform format from multiple reanalyses for point locations.

2   GlobSim enables efficient ensemble simulations at single or multiple points.

Compared to simulations forced by individual reanalyses, ensemble means often yielded better performance with reduced RMSE in addition to providing information about predictive uncertainty.



*Code availability.* GlobSim is developed as a Python package available at https://doi.org/10.5281/zenodo.3237258 as a GPL-3.0 project in the version published here and with documentation at http://github.com/geocryology/globsim. A docker container containing the required libraries is also available at https://hub.docker.com/r/geocryology/globsim.

*Data availability.* Observations are available from the Nordicana-D data repository (Gruber et al., 2018).

*Author contributions.* BC carried out this study by analyzing data, developing part of the GlobSim code and performing the simulations. XQ contributed to the development and design of GlobSim. NB contributed to the development and testing of GlobSim, supported the operation of GEOtop on the virtual machines, tested the initial GlobSim parameters and conducted the field measurements. ESJ contributed to the collection of field measurements, developed the terrain-type classifications and assisted with the selection of model parameters. SG conceived and guided the project, and designed the initial code of GlobSim. All authors contributed to the writing and editing of the manuscript.

*Disclaimer.* The authors declare that they have no competing interests.

*Acknowledgements.* The authors thank the Southern Ontario Smart Computing Innovation Platform (SOSCIP), Ontario Centres of Excellence (OCE), IBM, and Carleton Research Computer Services for their support of this project. The ESMF library was downloaded from https://www.earthsystemcog.org/projects/esmf/ (last access: 21 Feb 2019) and we acknowledge the support of Ryan O'Kuinghttons and Robert Oehmke in finding a good way to use it for efficient interpolation in GlobSim. Luis Padilla-Ramirez provided valuable testing,
feedback and bug fixing.



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
