# Peer review of "GlobSim (v1.0): Deriving meteorological time series for point locations from multiple global reanalyses"

_Geoscientific Model Development, 2019_

## Referee Comment (RC1) · Anonymous Referee #1 · 3 Aug 2019

1 General Comments This paper proposed by Cao et al. presents a new tool, GlobSim, to derive meteorological variables from multiple reanalyses (ERA-Interim, ERA5, JRA-55, MERRA-2) for ensemble simulation. The motivation and novelty of the paper–as stated by the authors–lie in the technical challenges which limit the ease of reanalysis data can be applied to models at site scale. As far as I know, a tool like GlobSim does not exist so far. The authors show the suitability of GlobSim via applying it in a large number (156 sites) of soil temperature simulation in permafrost-affected regions. I am very impressed by the strength of GlobSim, combined with GEOtop, in capturing fine-scale temperature variability due to local scales, such as snow and vegetation cover, soil moisture, and a peat layer. In general, the paper and tool are well written and described, this is an interesting study. Although it lacks additional scaling methods

and is limited to site scale, GlobSim would be a useful tool in modeling a number of land surface processes. I have two concerns. (i) Long-term simulation The authors showed the changes of permafrost temperature at different depths (0.1, 10, and 20 m) since 1980. However, only the upmost 12 m soil profile is shown in Figure 7. How did the author conduct a simulation at the depth of 20 m. Please clarify. I can understand this is provided as a demonstration of the utility of GlobSim for supporting long-term simulation. However, permafrost temperature change at a long-term scale is normally complex. This is because it is driven by both climate conditions (air temperature, precipitation as already considered by the authors) and related factors, such as such as soil moisture and vegetation. Unfortunately, the authors have not mentioned these at all. Given the description in Table 3, I assume the simulations present here is heavily simplified by ignoring such important processes. Please clarify. I suggest the authors, at least, discuss the potential influences on the temperature influences. (ii) Spatial interpolation A number of metrological variables from three fields are derived and processed (interpolation and scaling) in GlobSim. As authors mentioned, the pressure level is also interpolated or derived via 3D interpolation. However, some of the variables, such as air temperature, wind speed and direction, are available for both the surface analysis and pressure level analysis. In this context, what's the field (surface analysis or pressure level analysis) used here in permafrost simulation? More general, what is the selection strategy for such variables? This is important because different treating ways, 2D interpolation for surface analysis and 3D interpolation for pressure level analysis, would lead to different values for the same variable. This has been demonstrated by authors' another paper (Cao et al, 2017). 2 Specific comments P7, L8: Should it be term "scaling"?

P9, L31: change . . .types in in the area to . . .types in the area. . .

P10, Figure 3: Relative humidity of MERRA-2 is missing. Also consider adding label for each subplot as you've done for the other figures since you have eight figures here.

P11, Figure 4: what is the background, DEM or hillshade, please clarify. What does the

blue part in the upper right subplot? Seems the legend of Mine is not used, suggest delete it.

P12, L6: What do you mean Surface offset is defined is used here...? Please clarify.

P13, Table 2: Seems the minus symbol (-) for the units of thermal capacity (should be 106 J m-3 K-1) and thermal conductivity (should be W m-1 K-1) is missing

P14, Table 3: Similar with Table 3, the minus symbol in many units are missing, please double-check.

P14, Table 3: simulation depth is not sufficient for the long-term simulation, which exceeds 20 m, please clarify.

P16, L10–12: The last sentence of this paragraph is very unclear. Please clarify.

P22, L19: Should it be at a site scale or at the site level? Finally, I look forward to further development of GlobSim. References Cao, B., Gruber, S., and Zhang, T.: REDCAPP (v1.0): parameterizing valley inversions in air temperature data downscaled from reanalyses, Geoscientific Model Development, 10, 2905–2923.

---

## Referee Comment (RC2) · Anonymous Referee #2 · 5 Aug 2019

Reviews for "GlobSim (v1.0): Deriving meteorological time series for point locations from multiple global reanalyses" by Cao et al.

This study develops, describes and validates the GlobSim, which aims to downscale the gridded reanalysis atmosphere data to a single point scale, in order to drive models for single simulation research. Single simulation is generally important, e.g., for development of model and research on mechanisms. But usually forcing data is scare, particularly in high mountain or high latitude regions, resulting in that many simulation experiments can not be carried out in this regions. This study attempts to better use the reanalysis data to solve this issue. The topic is important. The study also contains large amount of work, well writing and clear organization. Generally, it has a potential for publication. I have several comments that is considered to improve the paper

1. My main concern is the validation. The paper develops the GlobSim that aims to output the better atmosphere forcing data. So, first, the output results (atmosphere data) should be validated to see whether the better atmosphere forcing data are produced. Then they can be used to forcing model and some validations are further performed to see whether simulated results are better, which in turn also have a strong demonstration of the better forcing data. Now, the study directly validated the simulated results. As we known, simulation performances are determined by both forcing data and models. In this case, better simulation performance may be caused by model rather than forcing data, and so not reaching the main target that forcing data are actually needed to be validated.

2. I also suggest that this study used the direct atmospheric forcing data in site-matched grid's reanalysis data (i.e., value in a simulation grid) to carry out a reference experiment, and then to compare with GlobSim results (single point) forced results. This comparison can really demonstrate the advantage of GlobSim.

3. Page4, Line2, a newer literature (Estimates of global surface hydrology and heat fluxes from the Community Land Model (CLM4.5) with four atmospheric forcing datasets. Journal of Hydrometeorology. 2016, 17, 2493-2510) is suitable for this discussion.

4. Table 2, the resolution of JRA-55 should be 1.125 rather than 0.56.

5. Page15, L11, remove the "."

---

## Short Comment (SC1) · Anonymous Referee #2 · 26 Aug 2019

**Response to Anonymous Referee #2**

The authors would like to thank the reviewer for the constructive feedback, and the thorough assessment of the manuscript. Below we provide a point-to-point response to each comment, reviewer comments are given in black, responses are given in blue. Additionally, we have included details of how we intend to address these changes in a

revised submission.

Reviews for "GlobSim (v1.0): Deriving meteorological time series for point locations from multiple global reanalyses" by Cao et al.

This study develops, describes and validates the GlobSim, which aims to downscale the gridded reanalysis atmosphere data to a single point scale, in order to drive models for single simulation research. Single simulation is generally important, e.g., for development of model and research on mechanisms. But usually forcing data is scare, particularly in high mountain or high latitude regions, resulting in that many simulation experiments can not be carried out in this regions. This study attempts to better use the reanalysis data to solve this issue. The topic is important. The study also contains large amount of work, well writing and clear organization. Generally, it has a potential for publication. I have several comments that is considered to improve the paper

1 My main concern is the validation. The paper develops the GlobSim that aims to output the better atmosphere forcing data. So, first, the output results (atmosphere data) should be validated to see whether the better atmosphere forcing data are produced. Then they can be used to forcing model and some validations are further performed to see whether simulated results are better, which in turn also have a strong demonstration of the better forcing data. Now, the study directly validated the simulated results. As we known, simulation performances are determined by both forcing data and models. In this case, better simulation performance may be caused by model rather than forcing data, and so not reaching the main target that forcing data are actually needed to be validated.
Response: The basic assumption of GlobSim is that, as we stated in section 4, *"Reanalysis products are carefully designed and tested before release. In addition, many studies have evaluated their performance by inter-comparison, by*

*comparison with observations (e.g., Jiang et al., 2015) and by applying them to model simulation (e.g., Albergel et al., 2018; Beck et al., 2019).".*

Fiddes J. and Gruber, S. (2014); SenGupta and Tarbonton (2016) have done the validation with meteorological data and, for individual realanysis. This kind of study demonstrated that using surface and pressure level information can help to provide better data at fine scale with relatively simple methods and requiring a full atmospheric model. These were done at a place that is particularly useful to make this point: mountains because they have strong fine-scale variability, and the great in-situ data available could support such detailed evaluation.

GlobSim now enables the application of such methods more broadly. We demonstrate its utility (also having multiple reanalyses) at a location where topography does not dominate microclimatology. Otherwise, we would be re-doing Fiddes J. and Gruber, S. (2014); Sen, A. and Tarbonton, G. (2016)'s work in some way and thus distract the reader from what is really new here. What remains is the need for a tool that allows to generate time series and, where required, interpolate spatially. We use interpolation in this manuscript because it allows us to compare reanalyses on differing grids for the same point location.

For these reasons, we inter-compared the GlobSim-derived meteorological variables (Fig. 3) in order to appreciate differences or detect blunders in conversion rather than directly comparing them with observations, although a detailed air temperature evaluation is present (Figure 8a, Table 5).

2 I also suggest that this study used the direct atmospheric forcing data in sitematched grid's reanalysis data (i.e., value in a simulation grid) to carry out a reference experiment, and then to compare with GlobSim results (single point) forced results. This comparison can really demonstrate the advantage of GlobSim.

Response: As we simplify ground locations that span an area wider than a typical reanalysis cell into a common center point, the comparison of this point with grid results will have little meaning. This study is the first step (v.1.0), GlobSim has not

yet implemented other (e.g., Fiddes J. and Gruber, S., 2014; Sen, A. and Tarbonton, G., 2016; Cao et al., 2017), although the upper-air information has already been included. We hope significant improvements between grid and GlobSim would be achieved via coupling a number of scaling methods in the future.

3 Page4, Line2, Page4, Line2, a newer literature (Estimates of global surface hydrology and heat fluxes from the Community Land Model (CLM4.5) with four atmospheric forcing datasets. Journal of Hydrometeorology. 2016, 17, 2493-2510) is suitable for this discussion.
Response: Thanks, the manuscript of Wang et al. (2016) will be added in the revised manuscript.

4 Table 2, the resolution of JRA-55 should be 1.125 rather than 0.56
Response: Yes, the resolution of reanalysis is 1.25° or about 150 km (Kobayashi et al., 2016.)

5 Page15, L11, remove the "."
Response: It will be deleted in the revised manuscript.

**References**

Albergel, C., Dutra, E., Munier, S., Calvet, J.-C., Munoz-Sabater, J., de Rosnay, P., and Balsamo, G.: ERA-5 and ERA-Interim driven ISBA land surface model simulations: which one performs better?, Hydrology and Earth System Sciences, 22, 3515–3532, https://doi.org/10.5194/hess-22-3515-2018, 2018.
Beck, H. E., Pan, M., Roy, T., Weedon, G. P., Pappenberger, F., van Dijk, A. I. J. M., Huffman, G. J., Adler, R. F., and Wood, E. F.: Daily evaluation of 26 precipitation datasets using Stage-IV gauge-radar data for the CONUS, Hydrology and Earth System Sciences, 23, 207–224, https://doi.org/10.5194/hess-23-207-2019, 2019.

Cao, B., Gruber, S., & Zhang, T.: REDCAPP (v1.0): parameterizing valley inversions in air temperature data downscaled from reanalyses. Geoscientific Model Development, 10(8): 2905–2923. https://doi.org/10.5194/gmd-10-2905-2017, 2017.

Fiddes, J., and Gruber, S.: TopoSCALE v.1.0: Downscaling gridded climate data in complex terrain. Geoscientific Model Development, 2014, 7(1): 387–405. https://doi.org/10.5194/gmd-7-387-2014, 2014.

Jiang, J. H., Su, H., and Zhai, Chengxing and Wu, Longtao and Minschwaner, Kenneth and Molod, Andrea M. and Tompkins, Adrian M.: An assessment of upper troposphere and lower stratosphere water vapor in MERRA, MERRA2, and ECMWF reanalyses using Aura MLS observations. Journal of Geophysical Research, 122: 11, 468–11, 485. https://doi.org/10.1002/2015JD023752, 2015.

Kobayashi, S., Ota, Y., Harada, Y., Ebita, A., Moriya, M., Onoda, H., Onogi, K., Kamahori, H., Kobayashi, C., Endo, H., Miyaoka, K., Takahashi, K.: The JRA-55 reanalysis: General specifications and basic characteristics. Journal of the Meteorological Society of Japan. Ser. II, 2016, 93(1): 5–48. https://doi.org/10.1371/journal.pone.0169061, 2016.

Sen, A., and Tarboton, G.: A tool for downscaling weather data from large-grid reanalysis products to finer spatial scales for distributed hydrological applications. Environmental Modelling & Software, 84(10): 350–69. https://doi.org/10.1016/j.envsoft.2016.06.014, 2016.

Wang, A., Zeng, X., Guo, D.: Estimates of Global Surface Hydrology and Heat Fluxes from the Community Land Model (CLM4.5) with Four Atmospheric Forcing Datasets. Journal of Hydrometeorology, 2016, 17(9): 2493–2510. https://doi.org/10.1175/JHM-D-16-0041.1, 2016.

---

## Short Comment (SC2) · Anonymous Referee #1 · 28 Aug 2019

**Anonymous Referee #1**

**Response to Anonymous Referee #1**

The authors would like to thank the reviewer for the constructive feedback, and the thorough assessment of the manuscript. Below we provide a point-to-point response to each comment, reviewer comments are given in black, responses are given in blue. Additionally, we have included details of how we intend to address these changes in a revised submission.

**General Comments**
This paper proposed by Cao et al. presents a new tool, GlobSim, to derive meteorological variables from multiple reanalyses (ERA-Interim, ERA5, JRA-55, MERRA-2) for ensemble simulation. The motivation and novelty of the paper–as stated by the authors–lie in the technical challenges which limit the ease of reanalysis data can be applied to models at site scale. As far as I know, a tool like GlobSim does not exist so far. The authors show the suitability of GlobSim via applying it in a large number (156 sites) of soil temperature simulation in permafrost-affected regions. I am very impressed by the strength of GlobSim, combined with GEOtop, in capturing fine-scale temperature variability due to local scales, such as snow and vegetation cover, soil moisture, and a peat layer. In general, the paper and tool are well written and described, this is an interesting study. Although it lacks additional scaling methods and is limited to site scale, GlobSim would be a useful tool in modeling a number of land surface processes. I have two concerns.

**(1) Long-term simulation**
The authors showed the changes of permafrost temperature at different depths

(0.1, 10, and 20 m) since 1980. However, only the upmost 12 m soil profile is shown in Figure 7. How did the author conduct a simulation at the depth of 20 m. Please clarify. I can understand this is provided as a demonstration of the utility of GlobSim for supporting long-term simulation. However, permafrost temperature change at a long-term scale is normally complex. This is because it is driven by both climate conditions (air temperature, precipitation as already considered by the authors) and related factors, such as such as soil moisture and vegetation. Unfortunately, the authors have not mentioned these at all. Given the description in Table 3, I assume the simulations present here is heavily simplified by ignoring such important processes. Please clarify. I suggest the authors, at least, discuss the potential influences on the temperature influences.

Response: The long-term simulation is conducted based on 60 soil layers with a total depth of 50 m. We will add detailed introduction (see below) of long-term simulation in section 4.5. Additionally, we will add *"In the long-term simulation, the layer of bedrock is extended to 50 m."* in the caption of Figure 7 to clarify. Regard to the potential influences of soil moisture and vegetation, we will discuss the uncertainties.

*"Reanalysis produces multi-decadal meteorological variables (Table 1), and this makes simulating long-term changes of land-surface processes possible. To demonstrate the utility of GlobSim for supporting long-term simulation, we conducted an additional deep ground temperature simulation from 1980 to 2017 for a single terrain type. The soil profile increased to 60 layers with a total depth of 50 m via expending the bedrock layer. The model was spun up by repeating the reanalysis of 01/1980–12/1984 100 times (500 years). To improve simulation efficiency, we simplified GEOtop simulation by assuming the vegetation and soil moisture is constant over time. This is warranted as we aim to simply demonstrate the potential for long-term application."*

**(2) Spatial interpolation**

A number of meteorological variables from three fields are derived and processed (interpolation and scaling) in GlobSim. As authors mentioned, the pressure level is also interpolated or derived via 3D interpolation. However, some of the variables, such as air temperature, wind speed and direction, are available for both the surface analysis and pressure level analysis. In this context, what's the field (surface analysis or pressure level analysis) used here in permafrost simulation? More general, what is the selection strategy for such variables? This is important because different treating ways, 2D interpolation for surface analysis and 3D interpolation for pressure level analysis, would lead to different values for the same variable. This has been demonstrated by authors' another paper (Cao et al, 2017).

Response: Yes, both 2D (for single level) and 3D (for multiple levels or pressure level) are available in GlobSim. Given the relatively flat topography of the study area, here, we used only the single-level surface analysis or 2D interpolation. However, for mountains, 3D interpolation is preferred in order to capture the strong spatial variability of lapse rate caused by topography. This combined with suitable scaling methods (e.g., Fiddes J. and Gruber, S., 2014; Sen, A. and Tarbonton, G., 2016; Cao et al., 2017) will make GlobSim also suitable in mountains in the future. In the section 4.5 model setting and parameters, we will revise to
*"Given the gentle topography and small test area, the time-series were derived from single level analysis for the geographic centre of measurement sites, only (Figure 4)"*. to clarify.

Additionally, in section 3.3.2 Spatial interpolation, we will add the suggestion of selection strategy
*"Although, the 3D interpolation combined surface and atmospheric information is not used here due to the relatively flat study area, this would be useful for further development of GlobSim by coupling additional scaling methods."*

**Specific comments**

- P7, L8: Should it be term "scaling"?
Response: Will be corrected in the revised manuscript.

- P9, L31: change ...types in in the area to ...types in the area...
Response: The repeat of *"in"* will be deleted in the revised manuscript.

- P10, Figure 3: Relative humidity of MERRA-2 is missing. Also consider adding label for each subplot as you've done for the other figures since you have eight figures here.
Response: The relative humidity of MERRA-2 will be added as well as the label.

- P11, Figure 4: what is the background, DEM or hillshade, please clarify. What does the blue part in the upper right subplot? Seems the legend of Mine is not used, suggest delete it.
Response: The background is an elevation tinted hillshade. Elevation is represented by different hues and the topography is accentuated using darker pixel values. In this region, the elevation does not change significantly so the topographic variation from the hillshade is most prominent. The blue shaded region represents the Northwest Territories - a territory of Canada. The mine icon is visible at the leftmost edge of the map below the latitude marking and corresponds to the Ekati diamond mine main camp, a major landmark in the region. The caption has been changed to clarify:

*"Figure 4 Study area and approximate location of measurement sites. Many point locations overlap at the scale of the map, so the positions of their markers have been dispersed to better illustrate the number and type of sites. The centroid of all measurement sites (red star) was used as the location for which GlobSim*
*reanalysis data were derived. The top right inset map shows the location of the study area (red square) within the Northwest Territories (blue region) of Canada. The elevation tinted hillshade basemap is provided by the Natural Resources Canada Federal Geospatial Platform Elevation Data Web Mapping Service and is derived from the Canadian Digital Elevation Model (CDEM). Waterbody outlines were obtained from the ArcGIS online data library."*

- P13, Table 2: Seems the minus symbol (-) for the units of thermal capacity (should be 106 J m-3 K-1) and thermal conductivity (should be W m-1 K-1) is missing

- P2, L25-29: P14, Table 3: Similar with Table 3, the minus symbol in many units are missing, please double-check.
  Response: These are the responses of two above comments. The P13 and P14 of previous manuscript were not in correct format, and many of the symbols in Table 2 and 3 are missing "-". They will be fixed in the revised manuscript.

- P14, Table 3: simulation depth is not sufficient for the long-term simulation, which exceeds 20 m, please clarify.
  Response: A detailed introduction of long-term simulation will be added in the revised manuscript. Please see our responses to your major comment 1. In Table 3, the column will be revised to "12/50", and a caption will be added: *"The simulation depth of 12 and 30 m was used for GST and long-term simulation, respectively."*

- P16, L10–12: The last sentence of this paragraph is very unclear. Please clarify.
  Response: The sentence will be changed to *"The ensemble mean, which had a daily RMSE of 1.87 ℃, outperformed all individual reanalyses during the measured period and reduced RMSE by 0.07–1.78 ℃ (Figure 8)"*

- P3, L6: Should it be at a site scale or at the site level?
  Response: In the revised manuscript, it will be changed to *"at a site scale"*

Finally, I look forward to further development of GlobSim.

**References**

Cao, B., Gruber, S., & Zhang, T.: REDCAPP (v1.0): parameterizing valley inversions in air temperature data downscaled from reanalyses. Geoscientific Model Development, 10(8): 2905–2923. https://doi.org/10.5194/gmd-10-2905-2017, 2017.

Fiddes, J., and Gruber, S.: TopoSCALE v.1.0: Downscaling gridded climate data in complex terrain. Geoscientific Model Development, 2014, 7(1): 387–405. https://doi.org/10.5194/gmd-7-387-2014, 2014.

Sen, A., and Tarboton, G.: A tool for downscaling weather data from large-grid reanalysis products to finer spatial scales for distributed hydrological applications. Environmental Modelling & Software, 84(10): 350–69. https://doi.org/10.1016/j.envsoft.2016.06.014, 2016.

---

## Author Comment (AC1) · 12 Sep 2019

**Author's Response to Editor's Comments on *"GlobSim (v1.0): Deriving meteorological time series for point locations from multiple global reanalyses"**

Bin Cao[1,2], Xiaojing Quan[2], Nicholas Brown[2], Emilie Stewart-Jones[2], Stephan Gruber[2]

[1]National Tibetan Plateau Data Center, Institute of Tibetan Plateau Research, Chinese Academy of Sciences, Beijing, 100101, China
[2]Department of Geography & Environmental Studies, Carleton University, Ottawa, ON, K1S 5B6, Canada

**Correspondence**: Stephan Gruber (stephan.gruber@carleton.ca)

The authors would like to thank the two reviewer for constructive feedback, and the thorough assessment of the manuscript. This final response combines both previous individual responses (reviewer comments are given in black, responses are given in blue) and includes the changes now made to the manuscript resubmitted.

**Response to Anonymous Referee #1**

**General Comments**

This paper proposed by Cao et al. presents a new tool, GlobSim, to derive meteorological variables from multiple reanalyses (ERA-Interim, ERA5, JRA-55, MERRA-2) for ensemble simulation. The motivation and novelty of the paper–as stated by the authors–lie in the technical challenges which limit the ease of reanalysis data can be applied to models at site scale. As far as I know, a tool like GlobSim does not exist so far. The authors show the suitability of GlobSim via applying it in a large number (156 sites) of soil temperature simulation in permafrost-affected regions. I am very impressed by the strength of GlobSim, combined with GEOtop, in capturing fine-scale temperature variability due to local scales, such as snow and vegetation cover, soil moisture, and a peat layer. In general, the paper and tool are well written and described, this is an interesting study. Although it lacks additional scaling methods and is limited to site scale, GlobSim would be a useful tool in modeling a number of land surface processes. I have two concerns.

(i) **Long-term simulation**

The authors showed the changes of permafrost temperature at different depths (0.1, 10, and 20 m) since 1980. However, only the upmost 12 m soil profile is shown in Figure 7. How did the author conduct a simulation at the depth of 20 m. Please clarify. I can understand this is provided as a demonstration of the utility of GlobSim for supporting long-term simulation. However, permafrost temperature change at a long-term scale is normally complex. This is because it is driven by both climate conditions (air temperature, precipitation as already considered by the authors) and related factors, such as such as soil moisture and vegetation. Unfortunately, the authors have not mentioned these at all. Given the description in Table 3, I assume the simulations present here is heavily simplified by ignoring such important processes. Please clarify. I suggest the authors, at least, discuss the potential influences on the temperature influences.

Response: The long-term simulation is conducted based on 60 soil layers with a total depth of 50 m. We added detailed introduction (see below) of long-term simulation in section 4.5. Additionally, we added *"In the long-term simulation, the layer of bedrock is extended to 50 m."* in the caption of Figure 7 to clarify. Regarding the potential influences of soil moisture and vegetation, we discussed the uncertainties.

*"Reanalysis produces multi-decadal meteorological variables (Table 1), and this makes simulating long-term changes of land-surface processes possible. To demonstrate the utility of GlobSim for supporting long-term simulation, we conducted an additional deep ground temperature simulation from 1980 to 2017 for a single terrain type. The soil profile increased to 60 layers with a total depth of 50 m via expending the bedrock layer. The model was spun up by repeating the reanalysis of 01/1980–12/1984 100 times (500 years). To improve simulation efficiency, we simplified GEOtop simulation by assuming the vegetation and soil moisture are constant over time although uncertainties are excepted. This is warranted as we aim to simply demonstrate the potential for long-term application."*

(ii) **Spatial interpolation**

A number of meteorological variables from three fields are derived and processed (interpolation and scaling)

in GlobSim. As authors mentioned, the pressure level is also interpolated or derived via 3D interpolation. However, some of the variables, such as air temperature, wind speed and direction, are available for both the surface analysis and pressure level analysis. In this context, what's the field (surface analysis or pressure level analysis) used here in permafrost simulation? More general, what is the selection strategy for such variables? This is important because different treating ways, 2D interpolation for surface analysis and 3D interpolation for pressure level analysis, would lead to different values for the same variable. This has been demonstrated by authors' another paper (Cao et al, 2017).

Response: Yes, both 2D (for single level) and 3D (for multiple levels or pressure level) are available in GlobSim. Given the relatively flat topography of the study area, here, we used only the single-level surface analysis or 2D interpolation. However, for mountains, 3D interpolation is preferred in order to capture the strong spatial variability of lapse rate caused by topography. This combined with suitable scaling methods (e.g., Fiddes J. and Gruber, S., 2014; Sen, A. and Tarbonton, G., 2016; Cao et al., 2017) will make GlobSim also suitable in mountains in the future. In the section 4.5 model setting and parameters, we revised to

*"Given the gentle topography and small test area, the time-series were derived from single level analysis for the geographic centre of measurement sites, only (Figure 4)".* to clarify.

Additionally, in section 3.3.2 Spatial interpolation, we added the suggestion of selection strategy

*"Although, the 3D interpolation combing surface and pressure level information are not demonstrated here due to the relatively flat study area, it will be useful for further development of GlobSim by integrating additional scaling methods."*

**Specific comments**

- P7, L8: Should it be term "scaling"?
  Response: Corrected.

- P9, L31: change ...types in in the area to ...types in the area...
  Response: The repeat of *"in"* was deleted.

- P10, Figure 3: Relative humidity of MERRA-2 is missing. Also consider adding label for each subplot as you've done for the other figures since you have eight figures here.
  Response: Figure 3 is revised as below.

- P11, Figure 4: what is the background, DEM or hillshade, please clarify. What does the blue part in the upper right subplot? Seems the legend of Mine is not used, suggest delete it.
  Response: The background is an elevation tinted hillshade. Elevation is represented by different hues and the topography is accentuated using darker pixel values. In this region, the elevation does not change significantly so the topographic variation from the hillshade is most prominent. The blue shaded region represents the Northwest Territories – a territory of Canada. The mine icon is visible at the leftmost edge of the map below the latitude marking and corresponds to the Ekati diamond mine main camp, a major landmark in the region. The caption has been changed to clarify:

  *"Figure 4 Study area and approximate location of measurement sites. Many point locations overlap at the scale of the map, so the positions of their markers have been dispersed to better illustrate the number and type of sites. The centroid of all measurement sites (red star) was used as the location for which GlobSim reanalysis data were derived. The top right inset map shows the location of the study area (red square) within the Northwest Territories (blue region) of Canada. The elevation tinted hillshade basemap is provided by the Natural Resources Canada Federal Geospatial Platform Elevation Data Web Mapping Service and is derived from the Canadian Digital Elevation Model (CDEM). Waterbody outlines were obtained from the ArcGIS online data library."*

- P13, Table 2: Seems the minus symbol (-) for the units of thermal capacity (should be 106 J m-3 K-1) and thermal conductivity (should be W m-1 K-1) is missing

- P2, L25-29: P14, Table 3: Similar with Table 3, the minus symbol in many units are missing, please double-check.
  Response: These are the responses of two above comments. The P13 and P14 of previous manuscript were not in correct format, and many of the symbols in Table 2 and 3 are missing "-". They were fixed in the revised manuscript.

- P14, Table 3: simulation depth is not sufficient for the long-term simulation, which exceeds 20 m, please clarify.
  Response: A detailed introduction of long-term simulation is added. Please see our responses to your major comment 1. In Table 3, the column is revised to "12/50", and a caption was added: *"The simulation depth of 12 and 50 m was used for GST and long-term simulation, respectively."*

[Figure]

Figure 3. Inter-comparisons of monthly variables derived from GlobSim which are used in GEOtop during the period 09/2015–08/2017. SW$_d$ and LW$_d$ is the surface downwelling shortwave flux in air and downwelling longwave flux in air, respectively.

- P16, L10–12: The last sentence of this paragraph is very unclear. Please clarify.
  Response: The sentence is changed to *"The ensemble mean, which had a daily RMSE of 1.87°C, outperformed all individual reanalyses during the measured period and reduced RMSE by 0.07–1.78°C (Figure 8)"*

- P3, L6: Should it be at a site scale or at the site level?
  Response: It is changed to *"at a site scale"*

Finally, I look forward to further development of GlobSim.

**Response to Anonymous Referee #2**

Reviews for "GlobSim (v1.0): Deriving meteorological time series for point locations from multiple global reanalyses" by Cao et al.

This study develops, describes and validates the GlobSim, which aims to downscale the gridded reanalysis atmosphere data to a single point scale, in order to drive models for single simulation research. Single simulation is generally important, e.g., for development of model and research on mechanisms. But usually forcing data is scare, particularly in high mountain or high latitude regions, resulting in that many simulation experiments can not be carried out in this regions. This study attempts to better use the reanalysis data to solve this issue. The topic is important. The study also

contains large amount of work, well writing and clear organization. Generally, it has a potential for publication. I have several comments that is considered to improve the paper

1 My main concern is the validation. The paper develops the GlobSim that aims to output the better atmosphere forcing data. So, first, the output results (atmosphere data) should be validated to see whether the better atmosphere forcing data are produced. Then they can be used to forcing model and some validations are further performed to see whether simulated results are better, which in turn also have a strong demonstration of the better forcing data. Now, the study directly validated the simulated results. As we known, simulation performances are determined by both forcing data and models. In this case, better simulation performance may be caused by model rather than forcing data, and so not reaching the main target that forcing data are actually needed to be validated.

Response: The basic assumption of GlobSim is that, as we stated in section 4, *"Reanalysis products are carefully designed and tested before release. In addition, many studies have evaluated their performance by intercomparison, by comparison with observations (e.g., Jiang et al., 2015) and by applying them to model simulation (e.g., Albergel et al., 2018; Beck et al.,2019)."*.

Fiddes J. and Gruber, S. (2014); SenGupta and Tarbonton (2016) have done the validation with meteorological data and, for individual realanysis. This kind of study demonstrated that using surface and pressure level information can help to provide better data at fine scale with relatively simple methods and requiring a full atmospheric model. These were done at a place that is particularly useful to make this point: mountains because they have strong fine-scale variability, and the great in-situ data available could support such detailed evaluation. GlobSim now enables the application of such methods more broadly. We demonstrate its utility (also having multiple reanalyses) at a location where topography does not dominate microclimatology. Otherwise, we would be re-doing Fiddes J. and Gruber, S. (2014); Sen, A. and Tarbonton, G., (2016)'s work in some way and thus distract the reader from what is really new here. What remains is the need for a tool that allows to generate time series and, where required, interpolate spatially. We use interpolation in this manuscript because it allows us to compare reanalyses on differing grids for the same point location.

For these reasons, we inter-compared the GlobSim-derived meteorological variables (Fig. 3) in order to appreciate differences or detect blunders in conversion rather than directly comparing them with observations, although a detailed air temperature evaluation is present (Figure 8a, Table 5).

2 I also suggest that this study used the direct atmospheric forcing data in sitematched grid's reanalysis data (i.e., value in a simulation grid) to carry out a reference experiment, and then to compare with GlobSim results (single point) forced results. This comparison can really demonstrate the advantage of GlobSim.

Response: As we simplify ground locations that span an area wider than a typical reanalysis cell into a common center point, the comparison of this point with grid results will have little meaning. This study is the first step (v.1.0), GlobSim has not yet implemented other (e.g., Fiddes J. and Gruber, S., 2014; Sen, A. and Tarbonton, G., 2016; Cao et al., 2017), although the upper-air information has already been included. We hope significant improvements between grid and GlobSim would be achieved via coupling a number of scaling methods in the future.

3 Page4, Line2, Page4, Line2, a newer literature (Estimates of global surface hydrology and heat fluxes from the Community Land Model (CLM4.5) with four atmospheric forcing datasets. Journal of Hydrometeorology. 2016, 17, 2493-2510) is suitable for this discussion.

Response: Thanks, the manuscript of Wang et al. (2016) is added in the revised manuscript.

4 Table 2, the resolution of JRA-55 should be 1.125 rather than 0.56

Response: Yes, the resolution of reanalysis is 1.25°or about 150 km Kobayashi et al., 2016.

5 Page15, L11, remove the "."

Response: It is deleted in the revised manuscript.

[revised manuscript text omitted]